# Duration of Immunity in Cattle to Lumpy Skin Disease Utilizing a Sheep Pox Vaccine

**DOI:** 10.3390/vetsci11040164

**Published:** 2024-04-05

**Authors:** Varduhi Hakobyan, Khachik Sargsyan, Hasmik Elbakyan, Vazgen Sargsyan, Tigran Markosyan, Gayane Chobanyan, Manvel Badalyan, Satenik Kharatyan

**Affiliations:** 1Scientific Center for Risk Assessment and Analysis in Food Safety Area, 107/2 Masis Highway, Shengavit, Yerevan 0071, Armenia; khachikvsargsyan@mail.ru (K.S.); elbakyan1959@mail.ru (H.E.); vazgen1986@mail.ru (V.S.); tigran79hm@yandex.ru (T.M.); gayanechobanyan6115@maill.ru (G.C.); satenik.kharatyan@gmail.com (S.K.); 2Chair of Biosciences and General Chemistry, Armenian National Agrarian University, 74 Teryan Street, Yerevan 0009, Armenia; badalyan.manvel@mail.ru

**Keywords:** lumpy skin disease, vaccination, heterologous vaccine, duration of immunity

## Abstract

**Simple Summary:**

Lumpy skin disease (LSD) is categorized by the World Organization for Animal Health (WOAH) as a notifiable disease that has a negative economic impact on the agricultural industry. To control this disease, homologous vaccines based on a live attenuated LSD virus and/or heterologous vaccines based on the sheep and goatpox viruses are used. We have previously reported that a sheeppox vaccine is safe in cattle and does not induce viremia or clinical symptoms, which can occur with homologous vaccines. Our initial studies with a sheeppox vaccine in cattle identified an 86% population immunity at 30 days, but information on the duration of immunity, to confirm the utility of yearly vaccination, was lacking. We have identified that immunity wanes rapidly in young cattle after receiving a sheep pox vaccine, which suggests they should receive a booster vaccination 4–6 months after their first vaccination, followed by yearly boosters using a sheep pox-based vaccine.

**Abstract:**

The transmission of lumpy skin disease (LSD) occurs through ticks, mosquitoes, and flies. The most effective way to combat LSD is to conduct large-scale vaccination, covering the entire cattle population with safe and effective vaccines, while introducing restrictions on the movement of livestock. The first and only LSD cases that occurred in Armenia happened in 2015,and they were controlled with the use of a once yearly heterologous sheep pox vaccine for cattle in high-risk areas. We have previously reported on the safety and immunogenicity of this vaccine in cattle, but information on the duration of immunity is lacking. Our aim was to determine the duration of immunity to the LSD virus (LSDV) in cattle when utilizing a heterologous sheep pox vaccine. We have evaluated antibodies in cattle blood prior to and post-vaccination (1, 6, and 11 months). We have utilized an enzyme-linked immunosorbent assay to follow the development and waning of LSDV antibodies in vaccinated cattle in two age groups: 1) young unvaccinated cattle ≤12 months of age and 2) adult cattle that had previously been vaccinated. Our results were consistent with our previous study in Armenia, showing a high level of population immunity, 80.0–83.3%, in both age groups at 1 month, with a significant (*p* = 0.001) drop for young cattle at 6 months. Previously vaccinated adult cattle showed a longer duration of immunity at 11 months for this heterologous sheep pox vaccine. Based on these data, we advise that young cattle receive an additional booster vaccination 4–6 months after their first vaccination, and then yearly vaccinations in high-risk areas.

## 1. Introduction

Lumpy skin disease (LSD)is caused by a contagious virus of the family *Poxviridae*, genus *Capripoxvirus* (CaPV); it mainly affects cattle and water buffalo and, to a lesser extent, sheep and goats. LSD is characterized by the formation of necrotizing skin nodes (tubercles); fever; generalized lymphadenitis; swelling of the extremities; and lesion development on the eyes, respiratory organs, and mucous membranes. The CaPV genus also includes sheep pox virus (SPPV) and goat pox virus (GTPV), which are serologically indistinguishable and can provide heterologous cross-protection [1,2,3,4,5]. All CaPVs are included in the World Organization for Animal Health (WOAH) list of diseases that are subject to mandatory WOAH notification [6,7,8].

LSD is transmitted to animals by blood-sucking vectors, including ticks and mosquitoes, and its spread also includes mechanical transmission by flies. Rapid spread and disease can occur in areas with environments rich in the relevant vectors [9,10,11,12,13,14,15], which include Armenia.

Due to an increase in the epizootic situation of LSD in cattle spanning between Africa, Southeast Europe, the Middle East, Asia, and Russia, measures (e.g., increased diagnostic testing, vaccination) are being taken to prevent the further spread of this economically important disease [16,17,18].

It is necessary to have highly effective diagnostic tests for the rapid identification of diseases and specific response plans in place to cease the spread of disease once identified. Vaccination can also be effective during an outbreak to control the spread or as a preventative control measure to prevent LSD incursion [19,20,21]. It has been suggested that mass vaccination would provide effective protection if more than 80% of susceptible animals were vaccinated [5,22] with vaccines that elicit high efficacy and immunogenicity and cause no harm to the vaccinee [23]. Additionally, it is important to have a long immunity duration to protect between vaccinations and minimize the schedule of revaccination to support animal owners in low-income situations.

Currently, there are limited data on the duration of immunity induced by different CaPV vaccines, which has been highlighted as a notable gap by the European Food Safety Authority (EFSA) [24]. As countries define their vaccination plans for different animals and diseases, the duration of immunity must be clear to make decisions on how often animals need to be revaccinated to provide protection. Many countries also need to evaluate which vaccines are currently available [25,26] and understand the cost to the farmers who will be responsible for compliance of use [27,28]. For active LSD prevention and control, it is recommended to use a homologous, live attenuated viral vaccine derived from the Neethling strain and/or a heterologous live attenuated viral vaccine from the CaPV strains obtained from SPPV and GTPV. The vaccines that are commonly used include (1) the homologous LSDV South African Neethling strain; (2) the KSGP O-240 strain, previously described as the Kenyan SPPV and GTPV strains, but confirmed to be LSDV [29]; and/or (3) the heterologous RM65 Jovivac strain (Yugoslavian sheep pox strain), a Romanian SPPV and Gorgan GTPV strain [5,20,29,30].

LSD was first identified in Armenia among cattle in December 2015. To assess the outbreak and prevent the disease from spreading, Armenia implemented a vaccination campaign, utilizing a heterologous sheep pox vaccine in cattle for protection against LSDV. High-risk areas were identified and susceptible cattle were vaccinated once yearly using the 10X Sheep Pox Cultural Dry™ vaccine (Federal Centre for Animal Health (ARRIAH)),and no noted adverse effects and development of a strong antibody response 30 days post-vaccination were reported [31,32].

To determine the duration of immunity following administration of a heterologous sheep pox virus vaccine, we evaluated LSDV antibodies at 1, 6, and 11 months post-vaccination in young-aged cattle that were vaccinated for the first time and adult cattle that were vaccinated more than two times.

## 2. Materials and Methods

### 2.1. Study Area

All the cattle were located in the Ararat Region in west–central Armenia, where vaccination and serosurveillance for LSDV currently occur. This region borders Turkey and Azerbaijan and is a high-risk zone for LSDV.

### 2.2. Sample Size

For this study, we conducted a small convenience sampling with random selection. Cattle were selected randomly from each home where there were young animals <12 months of age who had never been vaccinated or where there were adult cattle that had been previously vaccinated during the yearly campaign. The adult cattle had more than two previous yearly vaccinations, with the last vaccination occurring in the spring of 2021. The sampling and vaccination started in April and continued to June 2022 for both young and adult cattle. We selected 30 animals for each age group from 5 different farms. A total of 240 samples were examined.

### 2.3. Vaccination

The Sheep Pox Cultural Dry™ heterologous vaccine produced by ARRIAH was administered as previously described [32]. Observations of the cattle on the day of vaccination and throughout the sampling procedures were recorded to monitor for any signs of LSD or other infectious diseases.

### 2.4. Sample Collection

The Scientific Council of the Scientific Center for Risk Assessment and Analysis in Food Safety Area approved the protocol for this trial in January 2022.

Blood was collected from all cattle prior to vaccination and at 1, 6, and 11 months post-vaccination. All the cattle were identified with an ear tag in advance to ensure traceability throughout the study. Blood was collected from cattle in the field and transported within 24 h to the laboratory under cold chain conditions. In the laboratory, the blood samples were centrifuged to separate the serum, and the samples were stored at 4 °C for up to 7 days prior to testing.

### 2.5. Detection of Antibodies in Cattle

The serum samples were evaluated for LSDV antibodies before and after vaccination using the ID Screen^®^ Capripox Double Antigen Multi-Species enzyme-linked immunosorbent assay (ELISA) (IDVet, Grabels, France) according to the manufacturer’s instructions [33], and all the samples were tested in duplicate. The samples were considered to be positive when the % SP was ≥30%. This ELISA detects antibodies to LSDV, SPPV, and GTPV.

### 2.6. Statistical Analysis

ELISA titers for the animals up to 12 months of age and the adult animals were compared before vaccination and at 1, 6, and 11 months post-vaccination by using *t*-tests. A *p*-value < 0.05 was considered the threshold of significance for all statistical tests. The statistical analysis was performed using SAS 9.4 (SAS Institute, Cary, NC, USA).

## 3. Results

To determine the duration of immunity in cattle vaccinated with a heterologous vaccine against LSDV, we conducted a study on the blood serum of the cattle of the two groups: (1) young cattle up to 12 months of age that received the vaccine for the first time; and (2) adult cattle that had previously received yearly vaccinations. The serum samples were analyzed before vaccination and at 1, 6, and 11 months post-vaccination by using an ELISA. Throughout the entire study period, all the cattle remained healthy in appearance with no obvious side effects in presentation or at the vaccine injection site. This is in concurrence with our previous study, which followed the same vaccination parameters and found there was no increase in body temperature or deviation from the physiological norm [32].

The results of the ELISA data are shown in Table 1 and Table 2. Prior to vaccination, all the cattle in both groups had negative ELISA titers based on the percent seropositive (SP) cut-off value of <30%. One month after vaccination, antibodies to LSDV were found in the animals of both groups. In Group I (young cattle), which received their first vaccination, 80% showed positive antibody titers (% SP range 45.8–83.9). In Group II (adult cattle), which had received at least two annual vaccines prior, 83.3% were antibody positive (% SP range 51.7–94.3%).

At 6 months post-vaccination, we observed a sharp decline in both groups of cattle. In Group I, only 6.7% of cattle (n = 2) had % SP-positive antibody titers, 43.2 and 47.8, respectively. For Group II, 43.3% had positive antibody titers (% SP range 31.8–65.7%). Finally, at 11 months post-vaccination, no cattle were seropositive in Group I, with all samples recording a % SP <30. For animals in Group II, positive levels of antibodies were detected in only 6.7% of adult cattle (n = 2), with % SP antibody titers of 30.9 and 31.2, respectively.

While we had a small sample size, we did perform a statistical analysis comparing the two groups of cattle at the four time points (Table 3). Based on the results, there were two time points at which the results were significantly different between the two groups: before vaccination (*p*-value <0.0001) and at 6 months post-vaccination (*p*-value = 0.001). The differences were not significant at 1 month or 11 months post-vaccination and should be evaluated in a larger study size.

## 4. Discussion

LSD is a highly contagious disease caused by LSDV, which was first identified in Armenia in late 2015 [31]. After confirming LSDV in Armenia, the veterinary authorities implemented the vaccination of cattle in high-risk areas, followed by passive surveillance on susceptible cattle. Veterinary authorities are presented with significant challenges when selecting effective vaccines due to the variability of testing, quality, efficacy, and affordability to farmers [22,34]. Since 2016, the Republic of Armenia has administered a heterologous sheep pox vaccine to protect cattle from LSDV [31,32,35], but the duration of immunity following vaccination with the heterologous vaccine was unknown.

Vaccines are a key element in the fight against LSDV. Previously in Armenia, [32] we assessed the quality of vaccination by determining the presence of specific antibodies before and 30 days after vaccination using ELISA, and we determined that the population immunity in cattle was high at 86.1% (range 83.8–87.9%), with no observed adverse side effects. This study has identified a similar effectiveness, with population immunity ranging from 80.0 to 83.3% on day 30 after vaccination.

Studies on the attenuated Neethling LSDV vaccine have shown that immunity to LSDV develops from day 10 and reaches its maximum peak 21 days after vaccination, with a duration of post-vaccination immunity of 1 year [36]. However, the data available in the literature regarding the detection of antibodies after vaccination are varied, with limited information available on the relationship of antibody titer with protection. This is important, as even when low-to-no detectable antibodies were identified, animals could still be protected from disease.

Previous studies on the Neethling strain vaccine have shown that antibodies could still be detected at 30 weeks using the virus neutralization assay (VNT) [6]. Additional studies with the Neethling strain, evaluating CaPV-specific antibodies using the VNT, immunofluorescence test (IFAT), and the same commercial ELISA utilized in this study, post re-vaccination, have detected antibodies at 46–47 weeks post vaccination in 35.06% of cattle by VNT and 33.77% by IFAT and ELISA, showing a similarity in the results of all three tests [37]. Furthermore, the sensitivity and specificity of the ELISA to the VNT were reported at 91% and 87%, respectively [37]. Studies using a local strain called Ismailia, a live attenuated LSDV vaccine, have reported that all calves in the study were registered to have protective antibodies by both ELISA and serum neutralization test (SNT) at 30 weeks and were still registering protective antibodies at 40 weeks when investigated via the ELISA [38]. For the specific prevention of LSDV, a homologous live attenuated virus (LAV) vaccine from the Neethling virus strain is commonly used, and it induces long-lived immunity for 3 years [39]. Additionally, a heterologous LAV vaccine utilizing sheep pox virus has been shown to induce strong cross-immunity for 2 years [40].

A recent comparison of an LSDV LAV vaccine and an inactivated LSDV vaccine determined that the cattle elicited a robust immune response following the LSDV LAV vaccine, and that all animals remained antibody-positive for up to 18 months and were protected from LSDV challenge [41]. The inactivated LSDV vaccine showed a good initial immune response, but not all animals were protected from LSDV challenge at 12 months [41].

To successfully control the disease, it is necessary to understand both the effectiveness of vaccination and the duration of immunity after vaccination. As there is limited information regarding the duration of immunity of all commercially available LSDV vaccines, we sought to elucidate this issue by utilizing a heterologous sheep pox vaccine in cattle.

Based on our research data utilizing the ELISA, we found that before vaccination young cattle had significantly lower titers (*p*-value <0.0001), with a mean of 9.59 (SD = 8.74), than the adult cattle with a mean of 18.96 (SD = 7.18), although the results were still seronegative (SP < 30%) for both groups. The results at 1 month post-vaccination were not significant, but at 6 months post-vaccination they were again significant with a *p*<0.001, and then again not significant at 11 months post-vaccination (Table 3). Based on these results following cattle vaccination with a sheep pox vaccine, we suggest that for young cattle receiving their first vaccination, the duration of immunity is 6 months, and for animals that received an annual revaccination, the duration of immunity is 11 months, which is consistent with other similar studies [37,38].

This study has limitations in that we were unable to perform the VNT for comparison with our ELISA or the previous study’s VNT results, as we do not perform viral culture in our laboratory. We also do not have the capacity to perform in vivo animal challenge studies with LSDV to correlate the ELISA titers with protection. These additional studies are necessary to obtain a complete picture of using a heterologous sheep pox vaccine for LSDV. Additionally, we suggest additional research into the cellular factors provoked by heterologous vaccines that improve long-lasting immunity, as previous studies using an inactivated CaPV have shown that poxvirus immunity is both humoral and cell-mediated [42]. It has also been reported that not all animals seroconvert following vaccination, yet they incur full protection against LSDV challenge [38]. Therefore, while the singular evaluation of anti-LSDV antibodies may not confirm protection in vaccinated animals, it is a valuable first step to consider when applying the effectiveness and evaluation of vaccinations in the field [34].

## 5. Conclusions

Based on our initial studies, by vaccinating cattle with the ARRIAH sheep pox vaccine we can provide decision makers in Armenia, and other countries utilizing this vaccine, with the knowledge that young cattle should receive a booster vaccination 4–6 months after their first vaccination and then yearly in high-risk areas, and that continued yearly vaccination in adult cattle in high-risk areas of Armenia is recommended.

## Figures and Tables

**Table 1 vetsci-11-00164-t001:** ELISA antibody titers as a percentage of seropositive (SP) or seronegative (SN) results in cattle prior to vaccination and 1, 6, and 11 months post-vaccination with sheep pox vaccine against LSDV.

Group	Before Vaccination				1 Month Post-Vaccination				6 Months Post-Vaccination				11 Months Post-Vaccination			
	SN †		SP		SN		SP		SN		SP		SN		SP	
	N *	% ‡	n	%	n	%	n	%	n	%	n	%	n	%	n	%
Group I: Cattle up to 12 months of age	30	100	0	0	6	20	24	80	28	93	2	6.7	30	100	0	0
Group II: Adult cattle >12 months	30	100	0	0	5	17	25	83	17	57	13	43	28	93	2	6.7

† SN = % SP < 30%.* n = number of animals. ‡ % = percent total of animals either seronegative or seropositive.

**Table 2 vetsci-11-00164-t002:** Individual cattle ELISA antibody titers as % SP before and following vaccination using a heterologous sheep pox vaccine against LSDV. % SP < 30 is negative (values in the table are the average of the test performed in duplicate).

#	Cattle Up to 12 Months of Age (Group I)				Adult Cattle (Group II)			
	Before	1 Month PV *	6 Months PV	11 Months PV	Before	1 Month PV	6 Months PV	11 Months PV
1	9.8	45.8	21.0	15.1	21.6	79.2	28.9	12.1
2	7.6	75.3	28.9	18.4	28.9	93.2	58.3	28.9
3	−0.1	25.6	19.6	8.7	19.1	75.9	26.7	10.1
4	4.7	58.7	29.1	15.9	10.3	69.1	11.3	2.3
5	5.4	47.0	17.8	6.7	9.5	87.5	39.4	19.9
6	24.6	83.9	47.8	29.8	25.0	71.0	27.0	17.0
7	−2.1	23.1	15.4	5.1	11.1	28.9	21.0	19.8
8	4.0	59.3	21.8	4.7	26.7	89.1	40.0	20.4
9	19.9	68.9	24.0	20.9	8.9	29.4	25.9	24.9
10	17.3	47.8	28.8	20.1	18.2	69.9	35.2	21.9
11	24.0	61.3	25.7	25.0	26.5	28.9	21.0	5.2
12	17.1	69.9	29.1	18.3	19.9	70.5	39.9	10.4
13	1.1	28.7	26.1	3.2	21.0	28.7	25.9	3.8
14	3.2	58.3	11.3	3.9	17.1	59.1	24.1	6.6
15	23.2	67.5	27.3	22.8	29.6	94.3	65.7	31.2
16	11.7	81.1	29.0	18.1	28.7	81.2	49.1	17.1
17	0.7	54.7	17.9	5.4	9.9	68.0	32.6	13.9
18	23.2	79.6	43.2	22.9	13.2	59.9	31.8	7.1
19	9.8	49.8	21.4	17.5	19.0	69.7	29.7	11.1
20	0.9	29.5	18.4	5.3	18.6	27.5	14.0	9.3
21	1.3	75.1	26.0	12.8	7.9	68.1	24.8	8.4
22	−0.4	19.4	8.7	3.2	29.0	59.6	25.6	2.3
23	9.8	66.4	25.1	10.9	28.9	84.5	60.8	30.9
24	25.6	79.2	27.2	18.5	14.5	51.7	20.1	9.7
25	2.3	26.4	11.8	7.6	25.3	89.0	25.9	6.5
26	2.9	80.2	29.8	13.0	9.9	77.9	28.9	14.0
27	17.4	79.4	18.8	18.0	18.6	75.8	33.0	19.9
28	10.5	69.9	21.5	17.8	10.1	64.7	28.1	17.2
29	6.6	59.7	14.5	8.8	24.6	90.1	56.7	20.3
30	5.7	61.0	10.9	6.5	17.3	84.0	49.2	11.9

# Cattle number (Group I (n = 30), Group II (n = 30)). * Post-vaccination (PV).

**Table 3 vetsci-11-00164-t003:** Statistical comparison of young and adult cattle ELISA % SP titers by *t*-test.

Group	Before		1 Month PV *		6 Months PV		11 months PV	
	Mean (SD)	*p*-Value	Mean (SD)	*p*-Value	Mean (SD)	*p*-Value	Mean (SD)	*p*-Value
I	9.59 (8.74)	<0.0001	57.75 (19.59)	0.064	23.26 (8.65)	0.001	13.50 (7.41)	0.630
II	18.96 (7.18)		67.55 (20.65)		33.35 (13.76)		14.47 (8.14)	

* Post-vaccination (PV).

## Data Availability

The datasets analyzed during the current study are available from the corresponding author upon reasonable request.

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
