# Peer review of "Duration of Immunity in Cattle to Lumpy Skin Disease Utilizing a Sheep Pox Vaccine"

_vetsci, 2024, doi:10.3390/vetsci11040164_

Round 1

Reviewer 1 Report

Comments and Suggestions for Authors

This study investigated the duration of immunization with CaPV vaccine against LSDV infection in adult and young cattle. A total of 60 cattle were blood sampled before and at months 1, 6 and 11 after immunization and serum antibody titers were determined by ELISA in immunized cattle and appropriate recommendations for immunization were given. The manuscript was generally well written and easy to read. The subject matter was interesting and the analytical methods were of practical use. However, the manuscript needs careful editing. More detailed experimental data for repeatability and reproducibility validation should be provided.

The following specific issues should be addressed:

1.     The experimental design lacked neutralization experiments and in vivo animal challenge tests. There is also a lack of safety evaluation data,such as temperature, weight and blood parameters.

2.     Tables 2 and 3 are formatted in a different font than table 1, so please standardize the formatting, including fonts and alignment.

3.     The results of ELISA experiments need to be given as duplicates. It is recommended to use line graphs or bar graphs to better emphasize the trend.

Author Response

Dear Reviewer 1,

We greatly appreciate the time you have taken to provide us with valuable feedback and giving us the opportunity to improve our manuscript. We have responded to each of your comments below. Please let us know if we have addressed your comments or if there are improvements we can still make. All changes requested by reviewers have been highlighted in yellow in the updated manuscript.

Comment 1: The experimental design lacked neutralization experiments and in vivo animal challenge tests. There is also a lack of safety evaluation data,such as temperature, weight and blood parameters.

Response 1: We stated in the manuscript  that our study had limitations in that we were unable to perform VNT for comparison with our ELISA results  because we do not perform viral culture in our laboratory. We were also unable to conduct in vivo animal challenge studies with LSDV to correlate ELISA titers with protection. We fully agree with you  that these additional studies would be necessary to obtain a complete picture of the protection from heterologous sheeppox vaccine.

Regarding safety evaluation data, we indicated in our previous manuscript, “The serological response in cattle following administration of a heterologous sheep pox virus strain vaccine for protection from lumpy skin disease; Current situation in Armenia” /On the day of vaccination and throughout the study period, all cattle studied were healthy. Following vaccination, we observed no swelling at the injection site or any other side effects of vaccination. Overall, we observed no increase in body temperature or deviation from the physiological norm in all cattle. We have added an additional sentence in the results section to highlight this. In the field however, we are unable to take weight measurements of any cattle.

Comment 2: Tables 2 and 3 are formatted in a different font than table 1, so please standardize the formatting, including fonts and alignment.

Response 2: Thank you for your careful viewing and comment. All three tables are now in palatino linotype per the journals formatting.

Comment 3:   The results of ELISA experiments need to be given as duplicates. It is recommended to use line graphs or bar graphs to better emphasize the trend.

Response 3: ELISA results were conducted in duplicate and the tables show the average values of the duplicate tests. We have updated the wording in the table to reflect this.

Best Regards,

Varduhi Hakobyan

Reviewer 2 Report

Comments and Suggestions for Authors

Dear Authors

Please find some comments on the paper in the uploaded text.

The use of vaccines and the immune response to LSDV is complicated. Although this manuscript concluded that the heterologous vaccines are not ideal to use, many other factors play a role in determining this. Which diagnostic test is used, cellular immunity, protection, etc. as you eluted in the text. All these factors should be taken into consideration to come to a conclusion.   

Comments on the Quality of English Language

The English was good, but some areas should improve.

Author Response

Dear Reviewer 2,

We greatly appreciate the valuable feedback you have provided and allowing us the opportunity to improve our manuscript. We have responded to each of your comments below. Please let us know if we have addressed your comments or if there are improvements we can still make. All changes requested by reviewers have been highlighted in yellow in the updated manuscript.

Comment 1: The use of vaccines and the immune response to LSDV is complicated. Although this manuscript concluded that the heterologous vaccines are not ideal to use, many other factors play a role in determining this. Which diagnostic test is used, cellular immunity, protection, etc. as you eluted in the text. All these factors should be taken into consideration to come to a conclusion.  

Response 1: We stated in the manuscript  that our study had limitations in that we were unable to perform VNT for comparison with our ELISA results because we do not perform viral culture in our laboratory. We also did not evaluate cellular immunity and protection of animals after vaccination campaign and our research data are based only on the results of ELISA.

Comment 2:The humoral immune response against LSDV is very low and difficult to detect- it is good to test the presence of Abs but is the ultimate not to show protection? Thus challenging of animals

Response 2: Previous studies with an inactivated CaPV have shown that poxvirus immunity is both humoral and cell mediated. It has also been reported that not all animals seroconvert following vaccination, yet they incur full protection against a LSDV challenge .Therefore, while singular evaluation by ELISA of anti-LSDV antibodies may not confirm protection in vaccinated animals, they are a valuable first step to evaluate when applying the effectiveness and evaluation of vaccinations in the field.

Comment 3:.I think this is important-in the case of LSDV even very low titers of Abs will still protect the animal against challenge

Response 3: We completely agree with you, since in our Republic we carry out a single vaccination every year and, as was found in our studies, the titer of Abs is already falling by the 11th month, but we have not observed any outbreaks of the disease.

Comment 4:The use of the diagnostic techniques compared to the ELISA used in this study should be discussed-does this influence the outcome of Ab titers?

Response 4: As we hadn't been able to use different diagnostic techniques, we can only rely on literature data from other authors.

Comment 5 : Many of these studies are necessary to get the entire picture of using heterologous vaccines for LSDV.

Response 5: Yes, we completely agree with you. Singular evaluation of anti-LSDV antibodies may not confirm protection in vaccinated animals, they are a valuable first step to evaluate when applying the effectiveness and evaluation of vaccinations in the field.

Comment 6:This is an important outcome of LSDV vaccines and tests used to determine Ab status

Response 6: We completely agree with you and welcome additional studies to complete this picture.

Additionally, we have made all the suggested language changes throughout the document as Reviewer 2recommended. All changes were highlighted in yellow.

Best Regards,

Varduhi Hakobyan

Round 2

Reviewer 2 Report

Comments and Suggestions for Authors

All suggested changes have been made.

The added paragraph in the Discussion should improve.

E.g.

Antibodies could still be detected at 30 weeks post-vaccination using the Neethling strain [6]. Vaccination using a live attenuated LSDV vaccine containing a local strain called Ismailia detected antibodies at 40 weeks post-vaccination [37]. Studies evaluating CaPV-specific antibodies using the virus neutralization assay (VNT), immunofluorescence test and ELISA identified antibodies 47 weeks post-vaccination [38].

(Did all the tests perform the same, thus antibodies could be detected 47 weeks pv using all the different tests?)

Comments on the Quality of English Language

Although there are areas where this can improve, language use is good. 

Author Response

Dear Reviewer,

Thank you for your question that has allowed us to improve our manuscript and provide more detail on the value of the ELISA results. In response to the following in the discussion:

"Antibodies could still be detected at 30 weeks post-vaccination using the Neethling strain [6]. Vaccination using a live attenuated LSDV vaccine containing a local strain called Ismailia detected antibodies at 40 weeks post-vaccination [37]. Studies evaluating CaPV-specific antibodies using the virus neutralization assay (VNT), immunofluorescence test and ELISA identified antibodies 47 weeks post-vaccination [38]."

Comment: Did all the tests perform the same, thus antibodies could be detected 47 weeks pv using all the different tests?

Response: We have updated this paragraph (lines 193-203 highlighted) with additional details from the three referenced manuscripts to highlight the comparisons of results from multiple tests when evaluating antibody detection following vaccination.

"Previous studies with the Neethling strain vaccine have shown that antibodies could still be detected at 30 weeks using the virus neutralization assay (VNT) [6]. Additional studies with the Neethling strain, evaluating CaPV-specific antibodies using the VNT, immunofluorescence test (IFAT) and the same commercial ELISA utilized in this study, post re-vaccination, detected antibodies at 46-47 weeks post vaccination in 35.06% of cattle by VNT and 33.77% by IFAT and ELISA showing similarity of results by all three tests [38]. Furthermore, the sensitivity and specificity of the ELISA to the VNT was reported at 91% and 87% respectively [38]. Studies using a local strain called Ismailia live attenuated LSDV vaccine, reported that all calves in the study registered protective antibodies by both ELISA and serum neutralization test (SNT) at 30 weeks and were still registering protective antibodies at 40 weeks via the ELISA [37]."

Please let us know if this is in line with the additional details you recommend.

Best Regards,

Varduhi Hakobyan
